# Towards a Characterization of Late Talkers: The Developmental Profile of Children with Late Language Emergence through a Web-Based Communicative-Language Assessment

**DOI:** 10.3390/ijerph20021563

**Published:** 2023-01-14

**Authors:** Gianmatteo Farabolini, Maria Gabriella Ceravolo, Andrea Marini

**Affiliations:** 1Department of Experimental and Clinical Medicine, Università Politecnica delle Marche, 60126 Ancona, Italy; 2Department of Languages, Literatures, Communication, Education and Society, University of Udine, 33100 Udine, Italy

**Keywords:** language, neuropsychology of language, neurodevelopmental disorders, executive functions, working memory, attention, socio-conversational skills, late talkers

## Abstract

Children acquire language naturally, but there is variation in language acquisition patterns. Indeed, different internal and external variables play a role in acquiring language. However, there are open research questions about the contribution of different variables to language development. Moreover, with societal changes and due to the pandemic situation, there has been a growing interest in testing digitalization related to indirect language acquisition assessment. In this study, a web-based assessment survey was developed to (1) describe the relation between expressive vocabulary, Socio-Conversational Skills (SCS), gender, parental education, executive functions (EFs), and pretend play; (2) determine whether the survey can detect differences between late talkers (LTs) and children with typical language development; (3) identify children with “overall high” and “overall low” communicative-language scores to test the validity of expressive vocabulary as a main indicator to detect LTs. The parents of 108 Italian children (51 males) aged 24–36 months participated in the study. The results showed that expressive vocabulary correlates with measures of SCS (assertiveness and responsiveness) and is reliable in identifying LTs (d = 2.73). Furthermore, SCS and EFs contribute to better characterizing the developmental profile of children aged 24–36 months.

## 1. Introduction

Language is a complex cognitive function that is usually acquired spontaneously [1], even if with great interindividual variability [2]. Expressive vocabulary is one of the most used developmental markers to analyze early language development [3,4]. The variability in lexical developmental paths may depend on a range of different factors that are related to the development of the child’s expressive vocabulary [5], including bilingual exposure, Socio-Economic Status (SES), gender, or genetic background [1,6,7,8]. Typically, by 12 months of age, children can discriminate the phonemes of the language they are exposed to and begin to understand and produce their first words [9]. Around the age of 18 months, they reach a repertoire of 50 words. The gradual increase in their expressive vocabulary around 24 months is usually accompanied by the production of two-word utterances [10,11]. Indeed, by the age of three children should have a relatively rich mental lexicon that allows them to produce gradually more complex utterances [12].

Not all children follow this typical trend of linguistic development. Approximately 10% to 20% of children aged between 24 and 36 months may experience significant difficulties in lexical development even in absence of intellectual disability, brain lesions, hearing impairments, and/or cognitive difficulties associated with specific syndromes (e.g., Down Syndrome; Autism Spectrum Disorder, etc.) [13,14]. These children showing a late language emergence are usually termed late talkers (LTs; [15]) and are characterized by delayed expressive lexical skills that can be observed also in the receptive domain [16,17]. Most LTs will exhibit a significant lexical improvement after the age of three, allowing them to perform within normal limits on linguistic tasks. Nonetheless, some difficulties may persist in their daily communicative interactions [18,19,20,21].

The assessment of lexical skills and vocabulary size is one of the most used parameters to identify early language difficulties [3,4,22]. A recent scoping review focusing on both systematic reviews and primary studies showed that delayed receptive and/or expressive lexical skills are among the early predictors of Developmental Language Disorder (DLD), especially if they are accompanied by difficulties in syntactic comprehension or word combination up to 30 months [23].

However, the analysis of the expressive and/or receptive vocabulary alone is not enough to describe the developmental and functional profile of LTs [24]. Indeed, accumulating evidence suggests that an accurate evaluation of a child’s linguistic profile should also be extended to other linguistic domains [3] and should consider the potential impact of additional variables (e.g., SES) that in early childhood may predict later language outcomes [4]. 

### 1.1. Factors Related to Typical Language Acquisition

As mentioned earlier, language acquisition is affected by several factors which are often classified as internal or external factors. Internal factors are non-linguistic variables that are related to language development (e.g., cognitive development or pretend play). External factors are environmental variables (e.g., parental education). Some other variables might be considered in between these two categories. For example, Socio-Conversational Skills (SCS) can be considered as an internal factor since they are based on children’s abilities, but they are also necessarily related to their communicative environment.

Among the cognitive skills involved in language development and processing, Executive Functions (EFs) play a major role. They include inhibition, updating, and shifting [25]. In a study on toddlers younger than three, an association between two measures of EFs (i.e., working memory and inhibition) and both vocabulary and syntactic skills was found [26]. Furthermore, the performance on one task assessing shifting correlated with both expressive and receptive vocabulary and syntax. Other studies showed a relation between inhibition control and lexical [27] and both lexical and grammatical production [28] in children younger than three that may support the management of interference due to lexical competitors during stages of lexical selection [28,29].

Language development and functioning can also be affected by SCS and SES. SCS are often measured as responsive or assertive behaviors employed by children [30] in active communicative behaviors (e.g., actions and gestures), in language comprehension [31], or in the pragmatic use of language [32]. SES refers to the social and economic environment of the child [33,34] and can be assessed by looking at family income and/or parental education [35]. Children from lower SES backgrounds usually have poorer lexical scores compared to peers from higher SES already at 18 months [11,36,37]. Similarly, lexical and syntactic differences can be found between children from families with mid- and high-SES [7,38,39]. There is evidence suggesting that SES effects on language development may be moderated by maternal speech attitudes [40], supporting the hypothesis that at least some aspects of SES should be targeted when assessing language development [38].

As also highlighted by a systematic review, a further variable potentially affecting language development is the presence of symbolic (or pretend) play [41]. This is the child’s ability to make unconventional use of an object (e.g., a carrot used as a phone or a soldier who can fly). Contrasting theories support the hypothesis of a relationship between pretend play and language development, which can be considered separate [42] or strictly correlated for the cognitive symbolic abilities required for both language acquisition processes and pretend play activities [43]. Therefore, pretend play should also be considered when assessing both communicative and linguistic development [44].

### 1.2. Factors Related to Language Acquisition in Late Talkers

Because of their impact on language, the aforementioned factors (i.e., EFs, SCS, SES, and pretend play) should be considered when assessing language development in children. Indeed, the severity and persistence of Developmental Language Disorder (DLD) correlate with both working memory and interference control [45,46]. Furthermore, working memory was a significant predictor of language scores before 4 years of age [47] while children with lower lexical abilities performed worse than peers with average or better lexical abilities on tasks assessing inhibitory control [48].

Looking at SCS, LTs have lower assertive behaviors in communicative language than their peers with typical language development (TLD) [30]. Moreover, their assertive behaviors do not promote the continuation of the communicative interaction [49]. Interestingly, within the group of LTs, those with poorer receptive language scores may show more assertive behaviors, supporting the hypothesis that they need the caregiver’s input and help to bridge their linguistic gap [50]. Other authors found similar results that nonetheless supported different interpretations about the direction of such an association. Indeed, low scores in social abilities among LTs may be mainly due to their linguistic difficulties [51]. There is also evidence that communicative skills may explain the portions of variance of language scores in children with different onset and persistence of language delay, even if these are not the core variables in describing the developmental paths of LTs [31]. Finally, the interactional features of communication are often considered when delivering early indirect language interventions for children with late language emergence to support their language development [52,53]. However, a child’s SCS are not always directly assessed or trained.

As for SES as a risk factor potentially affecting language development, results are mixed. There is evidence supporting the hypothesis that SES is a risk factor for early language difficulties [4,54] and a predictive factor for DLD [55]. For example, paternal social status was a risk factor for late language emergence and maternal social status predicted language comprehension in a Finnish study [54]. On the other hand, other investigations did not report any significant association between SES and early language difficulties [11]. Moreover, SES measures seem to have a low [56] or mixed [57,58] predictive value for language difficulties. A meta-analysis assessed the impact of different predictors on language outcomes in LTs reporting that in a total of 1955 participants across 12 studies, expressive language was significantly associated with SES, even if with a small effect size [4]. Similarly, a recent scoping review revealed that SES may have a low impact as a risk factor in predicting DLD [23].

Finally, the potential association between pretend play and vocabulary scores in LTs may also be of critical interest. While it is common to investigate pretend play scores in children with autism spectrum disorders [59], it could be relevant to explore the relation between vocabulary and pretend play. If pretend play is associated with vocabulary, this may support the hypothesis that both linguistic and cognitive symbolization might share similar cognitive processes which are affected in LTs. On the contrary, a non-significant association among these variables could suggest an independent development among the symbolization skills, which could be more related to their communicative and cognitive developmental domains.

### 1.3. Web-Based Communicative-Language Assessment

A core advantage of using web-based communicative and linguistic assessment procedures is the possibility to conduct studies on very large samples from different geographical areas [60]. In an investigation performed in Norway [61], a research team digitally transcribed the MacArthur-Bates Communicative Development Inventories (MB-CDI; [22]), one of the most used tests to indirectly assess communicative and language development in early childhood. According to a recent systematic review, similar results were found in screening procedures’ accuracy across tools administered by parents or trained examiners (e.g., clinicians, nurses, and teachers; [62]). Another recent systematic review highlighted that parental indirect language assessment tools for screening procedures showed higher sensitivity and specificity than direct language assessment tools [63]. 

A similar web-based procedure was followed for Hebrew [64] and American Sign Language [65]. Recently, a web-based procedure to administer a communicative and linguistic assessment using the MB-CDI [60] has been developed. The procedure is available in different languages [64,66] but not in Italian. Furthermore, in these studies the data collected using this web-based MB-CDI were not compared to data pertaining to other domains such as EFs and SCS. To the best of our knowledge, this is the first study to use an indirect web-based language assessment tool remotely in Italian.

### 1.4. The Current Study

In line with the aforementioned premises, the current investigation planned to explore the following research questions: (Q1) describe the potential relation between expressive lexical skills of Italian-speaking children and parental education (a component of familial SES), EFs and SCS obtained via a web-based assessment survey; (Q2) determine whether the web-based assessment survey can detect differences between LTs and children with typical language development (TLD) and their respective characteristics; (Q3) identify children with “overall high” and “overall low” communicative-language scores and analyze their developmental profile to test the validity of expressive vocabulary as the main indicator to detect LTs. We hypothesized that (1) regarding Q1 parental education and measures of EF and SCS would correlate with lexical expressive abilities; (2) regarding Q2, the web-based assessment survey would allow us to detect LTs; (3) if the correlations between expressive lexical skills and parental education, measures of EFs and SCS are significant and if the web-based assessment survey allows us to detect LTs, expressive vocabulary is an accurate indicator to disentangle children with low and high communicative-language development scores.

## 2. Materials and Methods

### 2.1. Participants

One-hundred and eight monolingual Italian-speaking children aged between 24 and 36 months (age: mean 29 months; sex: 51 males, 47% of the total) and with no sensory, motor, or cognitive impairment participated in the study (see Table 1). 

All families were residents of different regions of Italy. In detail, they came from Ancona (N = 4), Arezzo (N = 1), Ascoli Piceno (N = 8), Benevento (N = 4), Bari (N = 3), Brindisi (N = 2), Cosenza (N = 1), Ferrara (N = 1), Fermo (N = 1), Latina (N = 1), Macerata (N = 26), Milano (N = 3), Napoli (N = 12), Novara (N = 3), Padova (N = 2), Parma (N = 1), Perugia (N = 7), Pesaro-Urbino (N = 1), Pisa (N = 1), Reggio Emilia (N = 1), Roma (N = 2), Siena (N = 1), Udine (N = 18), Vibo Valentia (N = 1), Vicenza (N = 2) and Viterbo (N = 1). Parental education was coded according to the United Nations’ ISCED 2011 classification (UNESCO Institute for Statistics, 2012), where education levels are ranked in nine levels, ranging from 0 (early childhood education) to 8 (doctoral degree). The means and standard deviation of demographic characteristics and descriptive statistics of the entire sample are reported in Table 1.

### 2.2. Procedure

Due to the pandemic situation, a digital rather than a paper assessment was carried out using Kobo toolbox, an open-access, and user-friendly survey software. We then created our web-based communicative-linguistic assessment survey. 

The web-based protocol was sent to potential candidates for the study with the help of our professional and personal network. Parents had access to the participation link from online announcements made by our Lab, cooperating institutions (associations, nursery schools, or social media accounts involved in early language development topics-related), and people who helped in disseminating the project (practitioners, pediatricians, scholars). No researchers or trained examiners have administered or supported parents while filling in the web-based assessment tool. 

Once the parents had filled in the survey and correctly submitted the form, we received in our Kobo Toolbox account the submitted data which we then exported to a spreadsheet. Data were collected in the European Union (EU), where research data collection and storage are governed by the Generalized Data Protection Regulation (GDPR) and its local instantiation in the legal system of the member states. Participant names were not collected, and birth dates were used to calculate the participants’ age in months. Our demographic questions did not contain any information that may be considered sensitive (e.g., information about children’s developmental disorders, linking data to IDs). Only the first author accessed data from the software and coded birth dates with no decimals in a new spreadsheet. Following these procedures, we collected data anonymously and we did not ask for participants’ privacy agreements.

### 2.3. The Survey

At the beginning of the survey, children’s demographic information (e.g., birth date, sex) was asked of parents. Before administering the indirect tests, we asked parents if their child had shown any motor development delay or was diagnosed with any neurodevelopmental (or neuropsychiatric) disorder. No parents reported any of these conditions. To assess the communicative-linguistic development of their children, parents received the digital version of the short form of the Italian adaptation of the MacArthur Bates—Communicative Developmental Inventory (MB-CDI; [22]) Words and Sentences [67]. This questionnaire asks parents to report the words spontaneously uttered by their children on a 100-words checklist. Notably, this survey allows clinicians to also gather information about the children’s morphosyntactic development with questions regarding, for example, the morphological complexity of the uttered words. Additionally, data for language comprehension and pretend play, through rating questions ranging from 0 to 2, were also obtained.

To assess the SCS of their children, parents received a specific survey (“Abilità Socio-Conversazionali del Bambino”, ASCB; [30]). It included 24 questions regarding the responsive and assertive behaviors of their children. Namely, for each item, parents were asked to rate the behavior of their children on a 5-point scale ranging from 0 to 4: 0 “mai” (*never*); 1 “quasi mai” (*rarely*); 2 “qualche volta” (*sometimes*); 3 “spesso” (*often*): 4 “sempre” (always).

To have an indirect assessment of the development of the EFs of their children, parents also compiled the paper version of the Italian adaptation [26] of the Behaviour Rating Inventory of Executive Functions—Preschool version (BRIEF-P; [68]). This questionnaire contains 63 items. For each item, parents were asked to report the frequency of their children’s behaviors in five developmental areas (i.e., inhibition, shift, working memory, planning/organization, and emotion regulation) on a 3-point scale: 1. “spesso” (*often*); 2. “qualche volta” (*sometimes*); 3. “mai” (*never*). From this test, we extracted data for each sub-test as well as a total EF score.

### 2.4. Statistical Analyses

We used SPSS to carry out the statistical analyses (see Appendix A for reproducible data). To answer Q1, the relation between the studied variables was investigated using Pearson product-moment correlation coefficient. To explore Q2, a series of between-group analyses were performed with Group (1. children with Typical Language Development; 2. Late talkers) as the independent variable and measures of parental education, SCS, pretend play, and EFs as dependent variables. The use of parametric or non-parametric analyses was decided on the results of Levene’s tests for equality of variances. Finally, a cluster analysis was performed to assess Q3; this statistical procedure splits the sample into two groups according to participants’ performance on target variables. It also allows analyzing if the two subgroups differ on the studied variables. Namely, a hierarchical method was first performed to define the number of clusters (k) for the K-means clusters. As a hierarchical method, we carried out an agglomerative clustering using Ward’s link that allows us to compute the sum of squared distances within clusters and to aggregate clusters with the minimum increase in the overall sum of squares. Following the Elbow rule, the k was set as the difference between the number of cases (our sample size) and the identification of the step in the agglomeration schedule where the “distance coefficients” makes a bigger jump. Our sample size is 108, while the bigger jump across distance coefficients is identified at stage 106. So, our non-hierarchical cluster analysis was run with a k set as 2, which means that we identified two clusters that split our samples in two groups that are relatively homogeneous within themselves and heterogeneous between each other. As above-mentioned, the main aim of this statistical analysis for the current research question was to explore significant differences for each variable among the two clusters of children see Appendix A for Code and Statistical analysis output).

## 3. Results

### 3.1. Q1—Are Expressive Lexical Skills Correlated with Parental Education and Measures of SCS, EF in the Overall Sample?

A point biserial correlation showed the absence of significant correlations between gender and expressive vocabulary (r = −0.081; *p* = 0.407). Pearson’s product-moment correlation analyses showed that the expressive vocabulary of the children included in the study did not correlate with pretend play (r = 0.046; *p* = 0.640), paternal (r = −0.154; *p* = 112) or maternal (r = −0.032; *p* = 0.745) education, or measures of EFs (r = 0.024; *p* = 0.807). However, expressive vocabulary had a medium positive correlation with the two measures of SCS (i.e., Assertiveness: r = 0.482; *p* < 0.01; Responsiveness: r = 0.440; *p* < 0.01) that showed a strong positive correlation with each other (r = 0.677; *p* < 0.01). 

A further inspection of the relation between the two measures of SCS and the other target variables showed that they did not correlate with age (Assertiveness: r = 0.132; *p* = 0.173; Responsiveness: r = 0.188; *p* = 0.052) nor with pretend play (Assertiveness: r = 0.003; *p* = 0.975; Responsiveness: r = 0.039; *p* = 0.689). However, they both correlated with the total score of EFs (Assertiveness: r = 0.358; *p* < 0.010; Responsiveness r = 0.398; *p* < 0.010) and the specific executive abilities: Inhibition (Assertiveness: r = 0.246; *p* < 0.010; Responsiveness: r = 0.316; *p* < 0.001); Shift (Assertiveness: r = 0.337; *p* < 0.010; Responsiveness: r = 0.251; *p* < 0.009); Working Memory (Assertiveness: r = 0.344; *p* < 0.010; Responsiveness: r = 0.410; *p* < 0.01); Emotion Regulation (Assertiveness: r = 0.344; *p* < 0.010; Responsiveness: r = 0.304; *p* < 0.001); Plan/Organization (Assertiveness: r = 0.204; *p* < 0.034; Responsiveness: r = 0.338; *p* < 0.001).

Finally, measures of EFs showed medium to strong correlations with each other. 

### 3.2. Q2—Can the Web-Based Assessment Survey Detect Differences between Late Talkers and Children with Typical Language Development?

LTs were identified if they had an MB-CDI lexical score below the 10th centile from the age of 24 months and/or no word combinations after the age of 30 months [24]. Overall, 14% of the total sample (i.e., 14 children, 10 males, 71%) aged between 24 and 34 months (mean = 29.30) showed late language emergence. Among these, 13 had an MB-CDI lexical score below the 10th centile, and one older than 30 months had both a low lexical score and no word combinations. No children older than 30 months had no combinations but a lexical score higher than the 10th centile.

The relation between gender and percentage of LTs or TLDs approached significance (χ^2^_(1, N = 108)_ = 3.782; *p* = 0.052) suggesting a trend toward a higher probability to be LT among males than females. Potential differences between children with TLD and LTs were assessed by performing a series of between-group statistical analyses. Levene’s test for equality of variances was significant for four dependent variables: expressive vocabulary (*p* < 0.001), the total score of EFs (*p* < 0.009), inhibition (*p* < 0.022), and emotional regulation (*p* < 0.001). For this reason, potential group-related differences in these four variables were assessed with a series of non-parametric Mann–Whitney tests. For all remaining variables, parametric analyses using *t*-tests were performed. The results of these analyses are presented in Table 2. 

As expected, the participants with TLD know and produce significantly more words than LTs (U = 1.316; *p* < 0.001). The two groups did not differ on paternal (t_(106)_ = −0.675; *p* = 0.501) or maternal (t_(106)_ = −1.099; *p* = 0.274) education. Similarly, no group-related differences were found in Pretend play (t_(106)_ = 0.037; *p* = 0.970). As for the two measures of SCS, children with TLD achieved higher scores on both Assertiveness (t_(106)_ = 5.149; *p* < 0.001) and Responsiveness (t_(106)_ = 5.172; *p* < 0.001). Finally, no group-related differences were found on the total score of EF (U = 0.871; *p* = 0.051) and on three of the executive measures: Inhibition (U = 841.50; *p* = 0.093), Emotion Regulation (U = 840.50; 0 = 0.094), Plan/Organization (t_(106)_ = 1.073; *p* = 0.286). However, on measures of Working Memory (t_(106)_ = 2.029; *p* < 0.045) and Shift (t_(106)_ = 2.805; *p* < 0.006), children with TLD were significantly better than LTs.

To better understand the profile of the two groups of participants, an additional series of Pearson product–moment correlation analyses were run independently for each of the two groups. These analyses showed that LTs and children with TLD had partially overlapping and partially different profiles. For both groups, the measure of expressive vocabulary did not correlate with pretend play (TLDs: r = 0.055; *p* = 0.596; LTs: r = 0.128; *p* = 0.663), paternal (TLDs: r = −0.156; *p* = 0.132; LTs: r = 231; *p* = 0.427), or maternal (TLDs: r = 0.033; *p* = 0.754; LTs: r = −0.305; *p* = 0.289) education. Similarly, in neither group significant correlations were found between expressive vocabulary and measures assessing EFs: Total EFs (TLDs: r = −0.172; *p* = 0.098; LTs: r = −0.080; *p* = 0.785), Inhibition (TLDs: r = −0.167; *p* = 0.107; LTs: r = 0.114; *p* = 0.699), Shift (TLDs: r = −0.160; *p* = 0.123; LTs: r = −0.140; *p* = 0.632), Emotion Regulation (TLDs: r = −0.180; *p* = 0.082; LTs: r = 0.140; *p* = 0.634), Working memory (TLDs: r = −0.074; *p* = 0.477; LTs: r = −0.372; *p* = 0.191), Planning/Organization (TLDs: r = −0.135; *p* = 0.193; LTs: r = −0.093; *p* = 0.751). However, measures of SCS showed different patterns of correlation between the two groups with significant correlations with expressive vocabulary only among children with TLD: Assertiveness (TLDs: r = 0.355; *p* < 0.001; LTs: r = −0.340; *p* = 0.234); Responsiveness (TLDs: r = 0.268; *p* < 0.009; LTs: r = 0.101; *p* = 0.732). Finally, a point biserial correlation showed the presence of significant correlations between gender and expressive vocabulary for LTs (r = 0.609; *p* < 0.021) but not for children with TLD (r = 0.045; *p* = 0.668).

### 3.3. Q3—What Is the Developmental Profile of Children with “Overall High” and “Overall Low” Communicative-Linguistic Performance?

The cluster analysis run through a non-hierarchical clustering method (K-means clustering) showed that the two groups reflect the presence of an “overall high performance” and an “overall low performance” group. An ANOVA was carried out to analyze the differences between the two clusters on each variable. Looking at the developmental profile of children with “overall high” and “overall low” communicative-linguistic development, the two groups differed significantly for expressive vocabulary [F(1,106) = 362.668; *p* < 0.001], assertiveness [F(1,106) = 15.497; *p* < 0.001], and responsiveness [F(1,106) = 13.656], while no further significant differences were found on the remaining variables: Pretend play [F(1,106) = 0.127; *p* = 0.722], Maternal education [F(1,106) = 0.037; *p* = 0.847]; Paternal education [F(1,106) = 1.789; *p* = 0.184], Total EF [F(1,106) = 0.33; *p* = 0.157]. Again, also in this case, no group-related differences were found in the specific executive skills: Inhibition [F(1,106) = 0.046; *p* = 0.831]; Shift [F(1,106) = 0.058; *p* = 0.811]; Emotional regulation [F(1,106) = 0.138; *p* = 0.711]; Working memory [F(1,106) = 0.195; *p* = 0.660]; Plan/organization [F(1,106) = 0.712; *p* = 0.401]. 

## 4. Discussion

The current investigation aimed at assessing the presence of potential associations between variables that are known to contribute to the cognitive and communicative-linguistic development of children aged between 24 and 36 months with a special focus on the developmental profile of LTs. Three major sets of analyses were conducted to answer the three research questions: the first included the whole sample (Q1); the second focused on the identification and characterization of late talkers (Q2); the third consisted of a cluster analysis to identify two potential subgroups according to their lexical expressive skills (Q3). The results will be discussed considering the available knowledge about lexical development.

As for Q1, when the whole sample of participants was considered the expressive vocabulary of the children included in the study did not correlate with parental education, pretend play, or EFs but showed significant correlations with measures of SCS. Notably, parental education was similar also between LTs and children with TLD and between children with “overall high” and “overall low” communicative-language scores. As in this study, parental education was selected as the only variable assessing SES (not including other variables such as family income and parental occupation), it is likely that this choice biased this finding, and that parental education alone may not be sufficient to account for SES effects on lexical development. The non-significant correlations between EFs and expressive vocabulary in the whole group does not support previous findings that had found associations between measures of EFs (e.g., updating as measured in terms of working memory, inhibitory control, and shifting) and measures of lexical and grammatical development [28,69,70]. It should be noted, however, that the available evidence on the potential relation between EFs and oral language production skills in preschoolers is still limited and controversial (e.g., [71]). This may be due to the wide interindividual variability observable in the linguistic and executive abilities of children under three years of age, the selection of different measures to assess EFs, and the young age of the children in this study. Indeed, it has been shown that the components of EFs [25] develop hierarchically in the child [72]. The development of EFs would become possible only after developing the ability to keep cognitive resources focused on a target for prolonged periods (sustained attention). This preliminary attentional ability would be joined, in successive stages, by the ability to maintain information in memory for the time necessary for its use (working memory), then by the ability to inhibit stimuli that are irrelevant to ongoing processing (inhibition), and finally by the ability to switch from one stimulus/task to another (attentional shifting). The abilities to plan/organize future actions and regulate emotions are likely developed later. A key point here concerns the fact that later-developing abilities would build on previously acquired skills. Indeed, performing an attentional shifting task requires the ability to inhibit irrelevant stimuli, hold this information in working memory, and focus the attentional resources in a sustained manner on the task for extended periods [73]. Consequently, difficulties in the development of early skills (e.g., sustained attention) could have significant repercussions on the development of later skills, such as working memory, the ability to inhibit interfering stimuli, or flexible switching from one task to another. Our evidence also showed a non-significant association between expressive vocabulary and pretend play, supporting the hypothesis that cognitive and linguistic symbolic skills reflect separate domains [42].

Interestingly, the measure of lexical expression correlated with the two measures of SCS that also correlated with each other and with EFs but not with pretend play or age. The significant correlation between expressive vocabulary and SCS subscales highlights the presence of a connection between SCS and lexical development supporting the idea that communicative skills and lexical development are deeply intertwined. Our evidence encourages targeting the child’s SCS in early language assessment and intervention. Furthermore, the significant correlations between SCS and EF scores highlight that the socio-communicative domain is related to more general cognitive development and suggest that they can influence each other. Further studies should highlight the direction of such associations by employing more participants and performing regression analyses. Of note, neither pretend play nor parental education correlated with expressive vocabulary or measures of SCS. This supports that the presence/absence of pretend play may not be a variable that significantly affects lexical development. 

As for Q2, the web-based survey allowed for the identification of a group of LTs. Their incidence in the whole sample was 14 percent, which is coherent with the estimates of the prevalence of language delay in the general population available in the current literature [13,14,15,46]. This is an important finding, as it supports the reliability of the web-based administration of such questionnaires. Interestingly, the analyses showed a trend of gender bias with males more prone to be LTs. This is coherent with previous findings suggesting that gender may be a predictive factor of language development in children under the age of three with males being more prone to be LTs and, potentially, to receive a diagnosis of DLD after the age of three [23]. Furthermore, LTs and children with TLD had only partially overlapping cognitive profiles. These two groups of participants were characterized by similar levels of parental education, similar scores assessing the presence of pretend play, and similar scores on three executive measures: inhibitory control, emotion regulation, and planning skills. However, they also differed on other skills. Namely, in line with previous studies [18], LTs scored lower on executive measures assessing updating (i.e., working memory) and the ability to shift between tasks or mental sets (mental set-shifting). These findings are particularly interesting as they do not support those by previous investigations showing that before the age of 3 the executive function system is still quite undifferentiated [74]. Interestingly, further exploration of the correlations between measures of working memory and those of shift showed that the two significantly showed a strong positive correlation in the group of LTs (r = 0.747; *p* < 0.001) and just a weak one among children with TLD (r = 0.365; *p* < 0.001). While supporting the hypothesis about stratification in executive skills [73], this finding highlights the role of working memory in the lexical difficulties observed in LTs. More specifically, previous investigations showed that a specific component of working memory (i.e., phonological working memory) plays a critical role in lexical acquisition and linguistic development [75]. In detail, it seems that lexical learning relies on the ability to process the perceived acoustic information, identify the corresponding phonemes, activate the relative semantic information and, eventually, map the phonological information onto the semantic one and store them in the mental lexicon in semantic declarative long-term memory [76]. A difficulty in the ability to keep the perceived phonological information in working memory might hamper the ability of children to store new words in memory. Indeed, in line with this hypothesis, phonological working memory difficulties are often observed in children with DLD [77,78] and, as is the case of our results, also in LTs [15,79]. Of note, LTs also had lower scores on tasks assessing social conversational skills (both Assertiveness and Responsiveness). This supports previous findings suggesting that in LTs this salient aspect of early pragmatic development is delayed [30]. Interestingly, measures of SCS showed different patterns of correlations between the two groups with significant correlations with expressive vocabulary only among children with TLD. This suggests that in LTs the association between social conversational skills and lexical development, normally present in children with typical development, may be anomalous. 

As for Q3, the cluster analysis allowed us to identify two groups in the whole sample based on their lexical expressive skills (i.e., those with “overall high performance” and one with “overall low performance”) that reflected the dichotomic division between LTs and children with TLD. Interestingly, the two groups significantly differed again in expressive vocabulary and social conversational skills. Expressive vocabulary is one of the two criteria used to identify LTs. The reduced expressive vocabulary found in children with “overall low” compared to those with “overall high” communicative and linguistic development confirms that the number of words pronounced by children is a reliable measure to identify those with high and low communicative-linguistic scores. Overall, then, our evidence suggests that the communicative dimension of language plays a core role in lexical development, as well as in the identification of linguistic difficulties. This evidence urges us to consider assertiveness and responsiveness when assessing language development. Further research should identify the contribution of SCS subscales in lexical development as well as their role in the identification of linguistic impairments.

To sum up, our evidence supports the need for including internal and external factors in early language assessment, to analyze language development through the association between different variables and expressive vocabulary and better define the functional profile of LTs. 

## 5. Limitations

This study has some limitations. First, it should be noted that the administration of the surveys was performed online using web-based research, while the assessment tool employed in the current study was originally developed and standardized for face-to-face language assessment. Notably, in a recent study [60], a web-based design was used as well but their participants filled in surveys under the supervision of practitioners, trained students, or researchers. For this reason, the reliability of our findings may have been affected by the absence of a researcher or a trained practitioner who supervised those who compiled the surveys. Future studies should carefully test the reliability of web-based administrations without supervision at providing unbiased results. For example, a pilot feasibility study might analyze the reliability of a web-based procedure administered to parents who fill in the survey next to a trained examiner (e.g., speech and language therapist) who might note any issues during the compilation. Once enhancements are made after the pilot, a comparison study between face-to-face and web-based administration could shed light on the reliability of the digital assessment tool.

As a second potential limitation, we highlight a potential selection bias. Even if participants were randomly contacted in different regions of Italy to promote our data collection, our sample might be mainly composed of a subgroup of the target population of parents who are informed and interested in communicative-language development topics. Therefore, these subgroups of parents might have enhanced and promoted communicative-language development in different ways compared to parents who have read the announcement of our research and decided not to participate in the study. Unfortunately, we were not able to target our data collection promotion to participants with low-SES backgrounds [60] to obtain more reliable information on our studied population.

## 6. Conclusions

This study used a web-based procedure for communicative, language, and cognitive assessment to characterize lexical and communicative development in children between 24 and 36 months. Overall, in line with previous findings [3,4,22,23], the results from this web-based assessment confirm that expressive vocabulary is a reliable measure to identify children with late language emergence. This also supports the possibility to adopt this methodology of administration in future studies and surveys. Furthermore, they support the usefulness of measures of SCS and EFs in better characterizing the communicative, linguistic, and cognitive profile of children between 24 and 36 months. Further research should address the role of different measures of SES in language development, as well as the value of early language assessments to predict subsequent and persistent language and neurodevelopmental difficulties.

## Figures and Tables

**Table 1 ijerph-20-01563-t001:** Demographic and cognitive characteristics of the children included in the study. Data are expressed as means and standard deviations, or percentages. Legend: SD—Standard Deviation; SCS: Socio-Conversational Skills; EFs—Executive Functions; na—Not available. * MB-CDI (MacArthur Bates Communicative Development Inventory) raw mean retrieved from normative data of children at the median age (29 months). ** MB-CDI percentage of parents reporting their children often do pretend play activities; the percentage is retrieved from scores of children at the median age (29 months). *** ASCB (Abilità Socio-Conversazionali del bambino) normative data are retrieved from 24-months children. **** BRIEF-P (Behaviour Rating Inventory of Executive Functions, Preschool version [26]) normative data are retrieved from 58 (27 males) 24–36 children.

		Sample (N = 108)		Normative Data Sample (N = 816)	
		24–36 Months		18–36 Months	
		Mean	SD	Mean	SD
	Age	29.29	3.73	28.83	5.1
MB-CDI	Maternal education	16.83	3.48	na	na
	Paternal education	15.30	3.75	na	na
	Expressive vocabulary	49.25° (64.97)	30.61° (28.82)	na (70) *	na (na)
	Pretend play	1.65	0.57	na	na
		69.44%	na	50% **	na
ASCB-SCS	Assertiveness ***	3.9	0.4	4.2	0.4
	Responsiveness	4.2	0.5	4.4	0.4
BRIEF-P-EFs	Total EF ****	153.60	18.34	89,34	17,53
	Inhibition	38.31	6.21	23.74	5.73
	Shift	26.37	3.29	13.56	9.04
	Emotional regulation	23.93	4.02	14.39	3.52
	Working memory	42.69	5.83	23.12	13.54
	Plan/organization	22.30	2.71	13.95	3.23
	Sex	M = 51 (47%)		M = 414 (50.7%)	

**Table 2 ijerph-20-01563-t002:** Characteristics of the two groups of participants identified through the analysis of their expressive vocabulary. Legend: TLD—(children with) Typical Development; LT—Late Talkers; EF—Executive Functions; SD—Standard Deviation. The asterisk (*) shows when the group-related difference was significant.

	TLD		LT	
	N = 94		N = 14	
	Mean	SD	Mean	SD
Age	29.29	3.70	29.30	4.06
Maternal education	16.69	3.56	17.79	2.81
Paternal education	15.20	3.74	15.93	3.83
Expressive vocabulary *	56.01	26.84	3.86	3.12
Assertiveness *	59.40	5.61	50.64	7.92
Responsiveness *	43.35	4.79	36.14	5.35
Pretend play	1.65	.56	1.64	0.63
Total EF	155.36	16.60	141.79	25.07
Inhibition	38.84	5.76	34.79	8.06
Shift *	26.70	2.97	24.14	4.44
Emotional regulation	24.29	3.58	21.50	5.83
Working memory *	43.13	5.43	39.79	7.67
Plan/organization	22.40	2.69	21.57	2.88
Sex	M = 41 (44%)		M = 10 (71%)	

## Data Availability

The data presented in this study are available in Appendix A.

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
