# Peer review of "Towards a Characterization of Late Talkers: The Developmental Profile of Children with Late Language Emergence through a Web-Based Communicative-Language Assessment"

_ijerph, 2023, doi:10.3390/ijerph20021563_

Round 1

Reviewer 1 Report

The introduction is in general well written, but I find it a bit fragmentary: it is difficult to get an overall picture. Each variable that is studied makes sense for the article on late talkers and the choice of each variable is appropriate. What I am missing is a more general rationale: what puts exactly these and only these variables together in a paper and in a study. This would need to be argued in the introduction to strengthen the rationale-maybe a theoretical model that guides the study might help?. It is stated that the role of variables that potentially contribute to explaining the LT profile are analysed. And this is correct, is well argued and discussed and is done well. However, why only these variables and not others? This choice was clinical or theoretical?. In other words, I suggest the autors to give a more coherent sense of the choice of variables. This will greatly strengthen the already very interesting work.

Please separate research questions from the introduction: You could add a further paragraph The current study.

I have a problem with the definition provided by the authors for the SES. Actually, I do not think that they have measured SES at all. In fact, what they have measured is parental education, which is a component of SES, but only one piece. This is a very reductive way to consider SES. I would suggest to the authors to call this variable Parental education rather than SES. The SES variable is multi-component, and includes at least 2 or 3 components to be defined as such. This is something that is considered in the discussion. But my suggestion refers to the fact that you should not call it SES at all!

In the section in which the descriptive statistics are reported, please insert where appropriate, the normative or average values otherwise the table is not very readable. For example the value corresponding to assertiveness: the average is 42. Is this an appropriate score? What is the maximum or desired score? The same applies to the other variables, such as responsiveness, pretend play, TOTAL EF, etc. Values cannot be understood without indicating the value to which they refer to.

As for the discussion, again as in the introduction I would suggest trying to make the information more coherent and compact. At the end of the discussion, the question remains: why were these variables studied together? I think clarifying this in the introduction will make it easier to discuss it at the end too, but it would be necessary to strengthen this aspect. The paper is well written but it is very difficult to capture the rationale.

Author Response

Reviewer 1

  1. The introduction is in general well written, but I find it a bit fragmentary: it is difficult to get an overall picture. Each variable that is studied makes sense for the article on late talkers and the choice of each variable is appropriate. What I am missing is a more general rationale: what puts exactly these and only these variables together in a paper and in a study. This would need to be argued in the introduction to strengthen the rationale-maybe a theoretical model that guides the study might help? It is stated that the role of variables that potentially contribute to explaining the LT profile are analysed. And this is correct, is well argued and discussed and is done well. However, why only these variables and not others? This choice was clinical or theoretical? In other words, I suggest the autors to give a more coherent sense of the choice of variables. This will greatly strengthen the already very interesting work. --> We thank the reviewer for this comment and the appreciation of our study. Indeed, the reviewer is right in pointing out that the former version of the Introduction was partially fragmentary. Now we have revised this part of the paper to provide a more general rationale. Furthermore, we have now clarified the reasons behind the choice of the variables. Namely, we have now added the following parts in the Introduction: “Expressive vocabulary is one of the most used developmental markers to analyze early language development [3,4]. The variability in lexical developmental paths may depend on a range of different factors which are related to the development of the child’s expressive vocabulary [5]”. We also added the following paragraph at the end of the Introduction: “However, the analysis of the expressive and/or receptive vocabulary alone is not enough to describe the developmental and functional profile of LTs [24]. Indeed, accumulating evidence suggests that an accurate evaluation of a child’s linguistic profile should be extended also to other linguistic domains [3] and should consider the potential impact of additional variables (e.g., SES) that in early childhood may predict later language outcomes [4]”. Furthermore, at the beginning of the “Factors related to typical language acquisition” section, we added the following passage:“ As mentioned earlier, language acquisition is affected by several factors which are often classified as internal or external factors. Internal factors are non-language variables which are related to language development (e.g., cognitive development or pretend play), while external factors are environmental factors not depending on the child’s characteristics (e.g., parental socio-economic status). Some other variables might be considered as in between these two categories. For example, socio-conversational skills can be deemed as an internal factor since they are based on children’s abilities, but they are also necessarily related to their communicative environment”. Then, regarding the potential role of SES in language acquisition, we added a sentence supported by a reference underlining why it is important to assess SES for a characterization of a child’s language development: “supporting the hypothesis that at least some aspects of SES should be targeted when assessing language development [38].” Finally, we also added a paragraph on the theoretical background supporting pretend play as a further variable to consider in language development assessment: “Contrasting theories support the hypothesis about the relationship between pretend play and language development, which can be considered separate [42] or strictly correlated for the cognitive symbolic abilities required for both language acquisition processes and pretend play activities [43].

  1. Please separate research questions from the introduction: You could add a further paragraph The current study. --> Following the reviewer’s suggestion now we have added a further paragraph at the end of the Introduction to outline the research questions.

  1. I have a problem with the definition provided by the authors for the SES. Actually, I do not think that they have measured SES at all. In fact, what they have measured is parental education, which is a component of SES, but only one piece. This is a very reductive way to consider SES. I would suggest to the authors to call this variable Parental education rather than SES. The SES variable is multi-component and includes at least 2 or 3 components to be defined as such. This is something that is considered in the discussion. But my suggestion refers to the fact that you should not call it SES at all! --> We thank the reviewer also for this useful comment. Indeed, at the end of the Introduction, when describing Q1 we had explicitly stated that we used one measure of SES, i.e., parental education. Consequently, we had consistently used SES in this sense in the article. However, we agree that this might be misleading. For this reason, now we have consistently renamed the variable “Parental education”

  1. In the section in which the descriptive statistics are reported, please insert where appropriate, the normative or average values otherwise the table is not very readable. For example, the value corresponding to assertiveness: the average is 42. Is this an appropriate score? What is the maximum or desired score? The same applies to the other variables, such as responsiveness, pretend play, TOTAL EF, etc. Values cannot be understood without indicating the value to which they refer to. --> We thank the reviewer also for this suggestion. Indeed, we have now implemented the table with the normative data from the original assessment tools when available. We highlight that not all means and SDs are available from the original tools. We also reported further information about normative data for each assessment tool in footnotes, below the table.

  1. As for the discussion, again as in the introduction I would suggest trying to make the information more coherent and compact. At the end of the discussion, the question remains: why were these variables studied together? I think clarifying this in the introduction will make it easier to discuss it at the end too, but it would be necessary to strengthen this aspect. The paper is well written but it is very difficult to capture the rationale. --> The Introduction section has now been extensively reshaped, at both the end of the first and at the beginning of the second section. We believe that now the need to consider both internal and external factors for an accurate assessment of language development in children is much clearer than in the earlier version of the paper. We thank the reviewer for pointing out the earlier weaknesses of the first draft. Following our evidence now at the beginning and at the end of the Discussion the importance of internal and external factors is explicitly stated.

Reviewer 2 Report

The biggest limitation of this study is the fact that parents filled on all the tests and that language conclusions are drawn on LT and TLD based on parents' expertise without previous training. Can authors comment on the reliability of results in the context of validation studies? The sensitivity of tests. It is suggested to validate the procedure in the sense that this “battery” should be given to parents facing the researchers face to face having explanations for each test plus the evaluation of the language domains by the speech and language professionals and psychologists. Then we would know much better about the language development of children assessed via parental information via the web.

-row 16- correct “IN”

-raw 33-34, what about the genetic background? Please add references.

-raw 159-163, please rewrite the sentence to be more understandable, so the reader does not need to “return again what is the result of 3 and 2.

“3) if the results of Q3 are like those for Q2, expressive vocabulary is an accurate indicator to disentangle children with low and high communicative-language development scores.”

-term “late talkers”and typical language development” are not used consistently in the manuscript text. Please introduce the acronym for each and then use the acronym in the text consistently (for example, row 282 versus 280..). As well term “socio-conversational skills” and acronym (SCS) please check the consistency in use in the manuscript text (for example row 217 versus 72..)

-It is suggested to add in Table 1 and Table 2 an explanation for each  variable what actually measures and what is the name of the test, maybe adding columns (for example, for assertiveness and responsiveness, SCS-ASCB [28] )

-It is suggested for the Results section not to use the paragraphs titles as with question mark, but in positive, as suggested, for example: “Results of the analysis on expressive lexical skills association with SES and measures of SCS, EF in the overall sample, “ etc.

-Raw 330-347 is related to the methodological part, can authors shift it somehow to the methodological section?

-          What this sentence means? “However, all the tasks used in this study were originally administered in person” It is somehow redundant because this was web-based study.

-          Raw 496- what does this means “newly developed web-based procedure? This is only the application of tests online, but nothing new was developed.

   The discussion somehow misses the information if other researchers were assessing the same via the web-based assessment by parents. 

Author Response

Reviewer 2

  1. The biggest limitation of this study is the fact that parents filled on all the tests and that language conclusions are drawn on LT and TLD based on parents' expertise without previous training. Can authors comment on the reliability of results in the context of validation studies? The sensitivity of tests. It is suggested to validate the procedure in the sense that this “battery” should be given to parents facing the researchers face to face having explanations for each test plus the evaluation of the language domains by the speech and language professionals and psychologists. Then we would know much better about the language development of children assessed via parental information via the web. --> We thank the reviewer for this comment. We have now added a paragraph in the “Web-based communicative-language assessment” section about reliability of reporting evidence from systematic reviews supporting indirect procedures for language assessment. The first paragraph of the “Limitations” section argued about the limitation of reliability of the web-based version since the assessment tool used in the current study had originally been developed for face-to-face language assessment. We have now implemented this paragraph.

  1. row 16- correct “IN” --> corrected.

  1. row 33-34, what about the genetic background? Please add references. --> We have now added also this information at the beginning of the Introduction.

  1. Row 159-163, please rewrite the sentence to be more understandable, so the reader does not need to “return again what is the result of 3 and 2. “3) if the results of Q3 are like those for Q2, expressive vocabulary is an accurate indicator to disentangle children with low and high communicative-language development scores.” --> Following the reviewer’s suggestion, We have now reformulated this sentence.

  1. term “late talkers”and typical language development” are not used consistently in the manuscript text. Please introduce the acronym for each and then use the acronym in the text consistently (for example, row 282 versus 280..). As well term “socio-conversational skills” and acronym (SCS) please check the consistency in use in the manuscript text (for example row 217 versus 72..) --> We have now consistently used the acronyms after their introduction.

  1. It is suggested to add in Table 1 and Table 2 an explanation for each variable what actually measures and what is the name of the test, maybe adding columns (for example, for assertiveness and responsiveness, SCS-ASCB [28] ) --> We have now added a column containing the name of the test and the domain assessed. We reported in a legend the acronym for the tests

  1. It is suggested for the Results section not to use the paragraphs titles as with question mark, but in positive, as suggested, for example: “Results of the analysis on expressive lexical skills association with SES and measures of SCS, EF in the overall sample, “ etc. --> As for this minor issue, we would like to keep it as it was in the first version of the paper. Indeed, in our opinion it is useful to remind the reader about the question that guided that specific analysis.

  1. Row 330-347 is related to the methodological part, can authors shift it somehow to the methodological section? --> We thank the reviewer for this observation. Now we have moved the description of the methodology for the cluster analysis to the “Statistical analyses” section.

  1. What this sentence means? “However, all the tasks used in this study were originally administered in person” It is somehow redundant because this was web-based study. --> Thank you, we have now removed this sentence.

  1. Row 496- what does this means “newly developed web-based procedure? This is only the application of tests online, but nothing new was developed. --> We have now deleted the words “newly developed”.

  1. The discussion somehow misses the information if other researchers were assessing the same via the web-based assessment by parents. --> In the procedure section, we have now added two sentences reporting “No researchers or trained examiners have administered or supported parents to fill in our web-based assessment tool. .” and “No other researcher had access to either the software or data..”.

Reviewer 3 Report

The paper concerns very important issue of how different variables contibute to the language development in children 24-36 months old. The authors present interesting results of their research and conclusions resulting from them. The paper is well written and can be published after a minor revision. I have only one suggestion, namely to replace the statements such as: "there is a lack of evidence about the contribution of different variables on language development" by e.g. "we continue research in the area of ..." or something like this (in the abstract and further parts of the paper) with suitable references. The more so that the authors refer to some papers containing such results.

Author Response

Reviewer 3

  1. The paper concerns very important issue of how different variables contibute to the language development in children 24-36 months old. The authors present interesting results of their research and conclusions resulting from them. The paper is well written and can be published after a minor revision. --> We thank the reviewer for the positive comments on our paper.

  1. I have only one suggestion, namely to replace the statements such as: "there is a lack of evidence about the contribution of different variables on language development" by e.g. "we continue research in the area of ..." or something like this (in the abstract and further parts of the paper) with suitable references. The more so that the authors refer to some papers containing such results. --> Following the reviewer’s suggestion, We have now modified the target statement in the abstract as well as through the whole manuscript.

Round 2

Reviewer 2 Report

-Please check this statement “To our knowledge this is the first study to use an indirect web-based language assessment tool remotely". Do you mean the first study in Italy in a web-based manner? It is suggested to rewrite the sentence and lower the statement.

-Table legends should be positioned below the table.

- Row 209 BRIEF-P (Behaviour Rating Inventory of Executive Functions) should include the reference

-Row 210, the full stop extra

-Did the web-based questionnaire include questions related to the diagnosis of neurodevelopmental and neurological disorders such as stuttering, autism etc? What were the exclusion criteria?

-Row 257-258, divide the Statistical analysis paragraph  from the previous text, there should be space

-Row 387-402, please rewrite the beginning of the Discussion it is a bit strange to start the discussion “As outlined.:” The reader knows the aim, and the discussion usually starts with the most important results of the study. There is no need to remind the reader of questions that can be found in the methodology.

-Row 505, what do you mean by “internal” and “external” factors? 

Author Response

-Please check this statement “To our knowledge this is the first study to use an indirect web-based language assessment tool remotely". Do you mean the first study in Italy in a web-based manner? It is suggested to rewrite the sentence and lower the statement. --> this sentence has now been modified to lower the claim.

-Table legends should be positioned below the table. --> Now we have controlled that all table legends are positioned below the tables.

- Row 209 BRIEF-P (Behaviour Rating Inventory of Executive Functions) should include the reference --> now the reference has been provided in this table legend.

-Row 210, the full stop extra --> the extra full stop has been deleted.

-Did the web-based questionnaire include questions related to the diagnosis of neurodevelopmental and neurological disorders such as stuttering, autism etc? What were the exclusion criteria? --> Yes, as reported in the Participants section, we asked for this information and no parents reported the presence of any sensorial, motor or cognitive disorders. We have now added a paragraph in The survey section.

-Row 257-258, divide the Statistical analysis paragraph  from the previous text, there should be space --> Done

-Row 387-402, please rewrite the beginning of the Discussion it is a bit strange to start the discussion “As outlined.:” The reader knows the aim, and the discussion usually starts with the most important results of the study. There is no need to remind the reader of questions that can be found in the methodology. --> The beginning of the Introduction had been rewritten like this as per request by Reviewer 1. However, we agree with Reviewer 2 that this reads a bit awkward. For this reason, we have modified it.

-Row 505, what do you mean by “internal” and “external” factors?  --> The definitions of internal and external factors is provided in the section labelled: “Factors related to typical language acquisition”.
